# Synthesis, Physicochemical, Labeling and In Vivo Characterization of ^44^Sc-Labeled DO3AM-NI as a Hypoxia-Sensitive PET Probe

**DOI:** 10.3390/ph15060666

**Published:** 2022-05-26

**Authors:** Dániel Szücs, Tibor Csupász, Judit P. Szabó, Adrienn Kis, Barbara Gyuricza, Viktória Arató, Viktória Forgács, Adrienn Vágner, Gábor Nagy, Ildikó Garai, Dezső Szikra, Imre Tóth, György Trencsényi, Gyula Tircsó, Anikó Fekete

**Affiliations:** 1Division of Nuclear Medicine and Translational Imaging, Department of Medical Imaging, Faculty of Medicine, University of Debrecen, Nagyerdei krt. 98., H-4032 Debrecen, Hungary; szucs.daniel@science.unideb.hu (D.S.); szabo.judit@med.unideb.hu (J.P.S.); kis.adrienn@med.unideb.hu (A.K.); gyuricza.barbara@med.unideb.hu (B.G.); arato.viktoria@med.unideb.hu (V.A.); forgacs.viktoria@med.unideb.hu (V.F.); garai.ildiko@med.unideb.hu (I.G.); szikra.dezso@med.unideb.hu (D.S.); trencsenyi.gyorgy@med.unideb.hu (G.T.); 2Department of Physical Chemistry, Faculty of Science and Technology, University of Debrecen, Egyetem tér 1., H-4032 Debrecen, Hungary; csupasz.tibor@science.unideb.hu (T.C.); imre.toth@science.unideb.hu (I.T.); gyula.tircso@science.unideb.hu (G.T.); 3Doctoral School of Chemistry, Faculty of Science and Technology, University of Debrecen, Egyetem tér 1., H-4032 Debrecen, Hungary; 4Doctoral School of Clinical Medicine, Faculty of Medicine, University of Debrecen, Nagyerdei krt. 98., H-4032 Debrecen, Hungary; 5Doctoral School of Pharmaceutical Sciences, Faculty of Pharmacy, University of Debrecen, Nagyerdei krt. 98., H-4032 Debrecen, Hungary; 6Scanomed Ltd., Nagyerdei krt. 98., H-4032 Debrecen, Hungary; vagner.adrienn@scanomed.hu (A.V.); nagy.gabor@scanomed.hu (G.N.)

**Keywords:** radiopharmaceutical, physicochemical study, scandium-44, radiochemistry, positron emission tomography, tumor hypoxia

## Abstract

Hypoxia promotes angiogenesis, which is crucial for tumor growth, and induces malignant progression and increases the therapeutic resistance. Positron emission tomography (PET) enables the detection of the hypoxic regions in tumors using 2-nitroimidazole-based radiopharmaceuticals. We describe here a physicochemical study of the Sc(DO3AM-NI) complex, which indicates: (a) relatively slow formation of the Sc(DO3AM-NI) chelate in acidic solution; (b) lower thermodynamic stability than the reference Sc(DOTA); (c) however, it is substantially more inert and consequently can be regarded as an excellent Sc-binder system. In addition, we report a comparison of ^44^Sc-labeled DO3AM-NI with its known ^68^Ga-labeled analog as a hypoxia PET probe. The in vivo and ex vivo biodistributions of ^44^Sc- and ^68^Ga-labeled DO3AM-NI in healthy and KB tumor-bearing SCID mice were examined 90 and 240 min after intravenous injection. No significant difference was found between the accumulation of ^44^Sc- and ^68^Ga-labeled DO3AM-NI in KB tumors. However, a significantly higher accumulation of [^68^Ga]Ga(DO3AM-NI) was found in liver, spleen, kidney, intestine, lung, heart and brain than for [^44^Sc]Sc(DO3AM-NI), leading to a lower tumor/background ratio. The tumor-to-muscle (T/M) ratio of [^44^Sc]Sc(DO3AM-NI) was approximately 10–15-fold higher than that of [^68^Ga]Ga(DO3AM-NI) at all time points. Thus, [^44^Sc]Sc(DO3AM-NI) allows the visualization of KB tumors with higher resolution, making it a promising hypoxia-specific PET radiotracer.

## 1. Introduction

Accurate tumor locating, staging and monitoring the impact of anticancer treatment are important issues in the fight against cancer. Positron emission tomography (PET) is a sensitive noninvasive imaging technique that can detect pathological processes at the molecular level in living systems. Hypoxia that develops during tumor growth induces cellular changes that result in malignant progression [1] and furthermore causes the following problems for anticancer treatments [2]. Oxygen deficiency triggers angiogenesis in solid tumors, but the formed vessels are abnormal; therefore, the delivery of the anticancer drugs to tumor cells is reduced [3]. Low-oxygenated cells are 2–3 times more resistant to ionizing radiation than healthy cells, which decreases the effectiveness of radiotherapy [4]. Consequently, the determination of the hypoxic region of tumors is key to achieving effective anticancer treatment. Because hypoxia is a significant challenge for oncologists, the development of radiopharmaceuticals capable of detecting low-oxygenated tumor cells is becoming increasingly important.

2-Nitroimidazole (NI) derivatives are excellent targeting vectors for imaging the hypoxic regions of tumors because they are reduced and entrapped by nitroreductase enzymes in hypoxic cells. In normal tissues, however, they are reoxidized and eliminated [5]. To date, some ^18^F-labeled radiopharmaceuticals containing a 2-nitroimidazole moiety were developed [6,7,8]. The two best known are [^18^F]-fluoromisonidazole ([^18^F]-FMISO) [9], which was the first PET radiopharmaceutical to be most widely evaluated in tumor hypoxia imaging, and [^18^F]-fluoroazomycin arabinoside ([^18^F]-FAZA) [10]. The lipophilic features of these PET agents provide diffusion into cells but cause slow clearance from healthy tissues, resulting in a low tumor-to-background ratio.

Faster clearance is achieved with radiolabeled chelator-conjugated 2-nitroimidazole derivatives [11]. In addition, the incorporation of radiometals into a suitable ligand offers a more straightforward avenue of radiolabeling than the nucleophilic radiofluorination, which often requires anhydrous conditions and a complex labeling procedure, resulting in low radiochemical yield. Therefore, radiometalation has recently become a favored process for peptides and other hydrophilic biomolecules [12,13,14,15,16,17]. Scandium-44 (t_1/2_ = 3.97 h, I = 94.27%, *E_mean_ (β+)* = 0.63 MeV) for radiolabeling has sparked our interest because of its excellent nuclear properties for PET imaging [12]. In addition, this positron emitter radionuclide has several advantages over the widely used gallium-68 (t_1/2_ = 68 min, I = 89%, *E_max_* (β+) = 1.92 MeV) [13]. The shorter-lived ^68^Ga radiometal can only be used in-house, while the relatively long half-life and large-scale production of the ^44^Sc by a cyclotron via ^44^Ca (p, n)^44^Sc reaction [14] allow transportation of ^44^Sc-labeled radiopharmaceuticals. Some studies have reported that radiolabeling with the ^44^Sc nuclide improves the pharmacokinetic features of the labeled complexes compared to ^68^Ga-labeled analogs [15]. Furthermore, the utilization of ^44^Sc-labeled radiotracer is also advantageous if a longer observation time is required for PET imaging (e.g., proteins, antibodies) [16]. In addition, beta emitter ^47^Sc is a valuable therapeutic match to the ^44^Sc nuclide for radiotheragnostic applications [12,17].

Hypoxia-selective uptake of ^68^Ga-labeled DO3AM-NI has been proved by Hoigebazar et al. [11]. Therefore, we have decided to achieve the radiolabeling of this DOTA-conjugated NI ligand (Figure 1) with the positron-emitting ^44^Sc isotope and compare the pharmacokinetic properties of this ^44^Sc-labeled complex with the known ^68^Ga-labeled version by in vivo PET imaging and ex vivo biodistribution. We have also conducted a detailed physicochemical study of the scandium (III) complex of DO3AM-NI to demonstrate its suitable thermodynamic stability, formation kinetics and kinetic inertness for in vivo use. The radiolabeling of 2-nitroimidizole derivatives with ^44^Sc isotope and the evaluation of the utility of the radiotracer for PET hypoxia imaging have not been previously reported.

## 2. Results and Discussion

### 2.1. Chemistry

#### 2.1.1. Synthesis

We carried out the synthesis of the DO3AM-NI ligand according to the literature’s methods. First, the preparation of 2-(2-nitro-imidazol-1-yl) ethanamine was performed by the slight modification of the procedure reported by Zha et al. [18]. Then, this compound was conjugated to 1,4,7,10-tetraazacyclododecane-1,4,7,10-tetraacetic acid (DOTA) by using the reaction conditions that were described by Hoigebazar et al. [11].

#### 2.1.2. Physicochemical Studies

The first step in the physicochemical studies of the Sc(III) chelate was to assess the rate of complex formation at an acidic pH, since based on equilibrium data published in the literature for Sc(III) complexes of DOTA, DTPA or AAZTA ligands one can expect that the formation of the (likely quite stable) Sc(DO3AM-NI) complex is expected at a low pH [19,20]. ^1^H and ^45^Sc NMR measurements performed at pH = 1.52 show that the complex formation reaches equilibrium in about twelve hours (Appendix A). This is promising for the labeling studies, yet it also underlines that the so-called “batch-method” must be applied in very acidic samples for the determination of the stability constant of the Sc(DO3AM-NI) chelate.

Equilibrium measurements were commenced by determining the protonation constants of the ligand, measured at 37 °C (*I* = 0.15 M NaCl) owing to the relevance of the chelate under in vivo conditions. The protonation constants obtained by fitting the pH-potentiometric data for the ligand DO3AM-NI are shown in Table 1 along with the literature data published for DOTA and certain DO3A-monoamides (DO3A-mono-N-buthylamide as well as that of a hypoxia-sensitive MRI probe reported by some of us a couple of years ago) [21]. Five protonation steps were observed by performing the titration in the pH range of 1.72–11.85 for the DO3AM-NI monoamide type ligand, two of which (characterized by high constants) describe the protonation of nitrogen atoms in the macrocycle (positioned *trans-* to each other), while the remaining log *K*_i_^H^ values (much lower) likely reflect the protonation of the carboxylates. The replacement of an acetate group of DOTA by an amide moiety results in a decrease in the protonation sites of the ligand, and as a consequence the overall basicity of the ligand (expressed as the sum of the protonation constants, log *β*_015_) was as expected, predicted from the data published in literature. What is more, the formation of a weak Na^+^ complex (originating from the ionic strength) competing with the first protonation process may be responsible of the lower value of the first protonation constant. Altogether, the basicity value is significantly lower than that for DOTA and comparable to/slightly lower than that of DO3AM-monoamides. The decrease in ligand basicity resulted in a decrease in the stability constants of the Ln(III) complexes, which is also expected for Sc(III) chelate.

The stability constant of the Sc(DO3AM-NI) complex was assessed by ^1^H and ^45^Sc-NMR studies using samples prepared in the 0.055–0.195 M acid concentration range. These data were supplemented by pH-potentiometric titration data obtained for the preformed complex to ensure the absence/presence of protonated and ternary hydroxydo species in solution at low and high pH, respectively. Absence of protonation and the presence of a ternary hydroxydo species formation with *K*_ScL•OH_ = 10.88(5) were detected at our experimental conditions. The NMR samples contained 3.32 mM ligand, ScCl_3_ and a constant ionic strength of 0.15 M NaCl. The samples were thermostated at 37 °C, and their ^1^H- and ^45^Sc-NMR spectra were recorded after 60 days. In order to ensure that the equilibrium in the samples was attained, the ^1^H and ^45^Sc-NMR measurements were repeated again 90 days later. The spectra were practically identical with the spectra recorded at 60 days (Appendix A). As expected, the thermodynamic stability constant of the Sc(DO3AM-NI) complex (log *K*_ScL_ = 22.36(4)) was considerably lower than those of Sc(III) complexes formed with DOTA, DTPA or AAZTA ligands (Table 1). A moderate decrease in terms of thermodynamic stability constants was already observed for the complexes of DO3A-monamides formed with Gd(III) ions in comparison to the stability of parent Gd(DOTA)^−^ chelate [21,22]. Despite the lower stability constant of the complex, the Sc(III) ion was complexed completely around pH = 2.0 (in a millimolar concentration range, see Figure 2). However, it might be misleading to directly compare the stability constants of the complexes formed by ligands of different basicity (as it depends on the basicity of the ligand); therefore, we calculated and compared the *p*Sc values (which could be treated as conditional constants) for the complexes shown in Table 1 by taking the protonation constants of ligands, the stability constants of the complex species present in solution and the hydrolysis of the Sc(III) ion (according to Ref. [23]: log *β*_Sc(OH)_ = −4.3, log *β*_Sc(OH)2_ = −9.7, log *β*_Sc(OH)3_ = −16.1, log *β*_Sc(OH)4_ = −26.0 and log *β*_Sc2(OH)4_ = −6.0) into account. These data confirmed the lower conditional stability of the Sc(DO3AM-NI) complex near the physiological pH as compared to those of Sc(AAZTA)^−^ and Sc(DOTA)^−^ chelates, yet the drop was much less pronounced compared to the difference of five to eight log units observed in the thermodynamic stability constant (i.e., significant part of the drop in the stability constant of Sc(DO3AM-NI) can be attributed to the differences in experimental conditions rather than the use of an amide moiety instead of a carboxylate metal binding unit causing a moderate drop in the conditional stability of the complex).

Safe in vivo applications of metal chelates require the utilization of very robust complexes characterized by high thermodynamic stability and inertness to avoid the transmetalation/transchelation reactions occurring with competing endogenous metal ions/ligands in the biological milieu. Since the stability of the Sc(DO3AM-NI) complex is lower than those of the complexes used as comparative benchmarks, we have probed its inertness. In most of the cases, the acid-assisted dissociation is recognized as a major dissociation path responsible for metal ion loss from the complexes of DOTA derivative ligands. Therefore, we have followed the dissociation rate of Sc(DO3AM-NI) complex in the presence of strong acid (1 M HCl) in order to gain some information about its inertness. The given study performed in duplicate gives rate constants of (1.55 ± 0.04) × 10^−6^ s^−1^ and (1.67 ± 0.05) × 10^−6^ s^−1^ (Appendix A), which is one-fourth of that observed for the Sc(DOTA)^-^ complex under identical conditions and several orders of magnitude smaller than the value previously found for the Sc(AAZTA)^-^ complex (0.1 M^−1^s^−1^) [19,20]. These data clearly show that the Sc(DO3AM-NI) chelate possesses superior inertness (slightly better than Sc(DOTA)^−^) and can be recommended for in vivo studies.

#### 2.1.3. Radiochemistry

The radiolabeling of the DO3AM-NI ligand with [^68^Ga]Ga^3+^ was previously described by Hoigebazar et al. [11]. For radiolabeling, we treated the [^68^Ga]GaCl_3_ solution, which was eluted from a ^68^Ge/^68^Ga-generator, with a post-processing method suggested by Eppard et al. [24] to reduce the eluate volume and remove metal tracer impurities. Then, the DO3AM-NI precursor was labeled with the conventional labeling procedure with good labeling yield (>90%). After radiolabeling, the radiotracer was purified by solid phase extraction (SPE) using a reversed phase C-18 SPE cartridge (SepPak C18 plus, Waters, Milford, USA). The radiochemical purity (RCP) of the purified labeled complex was analyzed by radio-HPLC and found to be >96%.

Cyclotron production of positron-emitting ^44^Sc was performed by proton irradiation of natural calcium targets via the ^44^Ca(p, n)^44^Sc reaction [14]. The ^44^Sc was separated from target materials and other metal-based impurities by the methods described by Happel et al. [25]. The radiolabeling with ^44^Sc was carried out by the modified version of the previously published procedure [15] with a high labeling yield (>93%). The radiotracer was purified by solid-phase extraction using a LiChrolute EN cartridge (Merck, Darmstadt, Germany). The RCP of the labeled product was determined by radio-HPLC, and it was over 95% (Appendix A).

Subsequently, the octanol/water partition coefficient (logP) of the labeled complexes was determined to be −3.89 for [^68^Ga]Ga(DO3AM-NI) and −2.59 for [^44^Sc]Sc(DO3AM-NI). The low logP values indicated that both radiotracers were hydrophilic. To assess the labeled compounds’ stability, the chelates were mixed with the solution of mouse plasma, Na_2_H_2_EDTA and oxalic acid, respectively. Aliquots were then taken at different time points (0, 60, 120 and 240 min) and injected into the radio-HPLC column, and the chromatograms were analyzed. After 240 min, the RCP of the samples was above 93% in all three cases. Appendix A shows the radio-HPLC chromatogram of the serum stability test at 240 min. These results show that both radioactive tracers possess high stability under the conditions studied.

### 2.2. In Vivo and Ex Vivo Studies

The biodistribution of ^44^Sc- and ^68^Ga-labeled DO3AM-NI was investigated using healthy control and KB tumor-bearing SCID mice. PET/MRI imaging and ex vivo studies were performed 90 and 240 min after intravenous injection of ^44^Sc- or ^68^Ga-labeled DO3AM-NI. Representative coronal PET/MRI images of a healthy control mouse are shown in Figure 3. By the qualitative analysis of the PET/MRI images, the kidneys and the bladder with urine were clearly visualized using both radiotracers at each investigated time point; however, in other abdominal and thoracic organs, low uptake was observed. After 90 min of incubation time, a relatively high accumulation was observed in the liver using [^68^Ga]Ga(DO3AM-NI) (Figure 3B). After the quantitative SUV analysis of the in vivo PET/MRI images, we found significantly (*p* ≤ 0.01) higher [^68^Ga]Ga(DO3AM-NI) uptake in the liver, spleen, kidney, intestines, lung, heart and brain than that of the [^44^Sc]Sc(DO3AM-NI) accumulation in the same organs 90 and 240 min after tracer injection (Figure 3E,F).

The results of the quantitative SUV analysis are correlated well with the ex vivo data shown in Table 2. Ex vivo biodistribution studies were carried out 90 min and 240 min after intravenous injection of ^44^Sc- or ^68^Ga-labeled DO3AM-NI, and the accumulated activities of the organs and tissues were determined using a gamma counter. By the quantitative analysis of the ex vivo data, significant differences were found between the %ID/g values when ^44^Sc- and ^68^Ga-labeled DO3AM-NI accumulations were compared. Significantly (*p* ≤ 0.01) lower %ID/g uptake values were measured in all the selected organs using [^44^Sc]Sc(DO3AM-NI) than that of the [^68^Ga]Ga(DO3AM-NI) at each investigated time point. Furthermore, notable accumulation was observed in the kidneys and urine using both radiotracers (Table 2), as it was expected from the log*P* values confirming that these radiolabeled complexes were excreted by the kidneys. The results of biodistribution studies correlated well with the ones obtained formerly for ^68^Ga-labeled nitroimidazole derivatives [26,27], which also showed low uptake and accumulation in abdominal and thoracic organs and were mainly excreted through the kidney. In contrast, e.g., [^18^F]fluorine- [28], [^131^I]iodine- [29], [^99m^Tc]technetium- [30] and [^64^Cu]copper-labeled nitroimidazole-based probes [31], due to their chemical properties, were mainly metabolized via the liver, and the large abdominal accumulation impaired the evaluation of abdominal tumors in the PET images. The ^44^Sc- and ^68^Ga-labeled nitroimidazole derivatives had more favorable properties in terms of the reporting and evaluation of PET images and due to their rapid elimination through the urinary system.

The tumor-specific accumulation of [^44^Sc]Sc(DO3AM-NI) and [^68^Ga]Ga(DO3AM-NI) was investigated by PET/MRI imaging 90 and 240 min after radiotracer injection using KB tumor-bearing mice. The representative decay-corrected images are shown in Figure 4. The subcutaneously growing KB tumors were clearly visualized using [^44^Sc]Sc(DO3AM-NI) 90 and 240 min post injection with relatively higher SUVmean and SUVmax values (SUVmean: 1.46 ± 0.32 and SUVmax: 2.35 ± 0.47 at 90 min and SUVmean: 0.67 ± 0.03 and SUVmax: 1.55 ± 0.28 at 240 min). Lower accumulation was found in the same KB tumors by using the ^68^Ga-labeled molecule (SUVmean: 0.79 ± 0.10, SUVmax: 1.94 ± 0.40 at 90 min and SUVmean: 0.51 ± 0.09 and SUVmax: 1.23 ± 0.10 at 240 min); however, this difference between the radiotracers was not significant (Figure 4E). In addition, 240 min after the injection of [^44^Sc]Sc(DO3AM-NI), the tumor-to-muscle ratios (T/M SUVmean: 182.67 ± 23.50, T/M SUVmax: 129.68 ± 26.11) showed significantly (*p* ≤ 0.01) higher values than that of [^68^Ga]Ga(DO3AM-NI) (T/M SUVmean: 85.66 ± 6.06, T/M SUVmax: 41.72 ± 13.61). In contrast, no significant differences were found between the T/M SUV values of the two radiotracers 90 min post injection (Figure 4F).

Overall, we found that the subcutaneously growing KB tumors were clearly visualized with excellent image contrast and higher SUVs, and T/M ratios were found by using the ^44^Sc-labeled molecule. Similar differences were found between the ^68^Ga- and ^44^Sc-labeled DOTA- and NODAGA-RGD molecules [32] or between the ^68^Ga- and ^44^Sc-labeled DOTA-NAPamide probes [15], when the tumor uptake and tumor-to-background ratios were investigated using in vivo tumor models, and higher accumulation was found using the ^44^Sc-labeled molecules. The difference between the ^44^Sc- and ^68^Ga-labeled DO3AM-NI was also due to the chemical properties and its longer half-life of ^44^Sc-DO3AM-labeled molecule.

For the assessment of the accumulation in the tumors of [^44^Sc]Sc(DO3AM-NI) and [^68^Ga]Ga(DO3AM-NI), ex vivo biodistribution studies were performed 90 and 240 min post injection using KB tumor xenografts. Table 3 demonstrates that 90 and 240 min after tracer injection no significant differences were found between the ^44^Sc- and ^68^Ga-labeled DO3AM-NI accumulations in KB tumors; however, the [^44^Sc]Sc(DO3AM-NI) uptake was relatively higher at each investigated time point. In contrast, when the tumor-to-muscle ratios (T/M) were calculated we found that the T/M ratio of [^44^Sc]Sc(DO3AM-NI) was approximately 10–15-fold higher at each time point than that of the T/M ratio of [^68^Ga]Ga(DO3AM-NI), and this difference was significant (*p* ≤ 0.01).

## 3. Materials and Methods

### 3.1. General

All reagents and solvents were obtained from commercial suppliers and used without further purification. 1,4,7,10-Tetraazacyclododecane-1,4,7,10-tetraacetic acid (DOTA) was purchased from ChemaTech (Dijon, France). All other reagents were purchased from Sigma-Aldrich, Saint Lousis, USA. TLC was performed on Kieselgel 60 F254 (Merck, Kenilworth, NJ, USA) with detection by UV detector. The ^1^H (400 MHz) and ^13^C NMR (128 MHz) spectra were recorded with Bruker DRX-400 spectrometers. Internal references: TMS (0.000 ppm for ^1^H), CDCl_3_ (77.00 ppm for ^13^C in organic solution). LC–MS was performed using a Waters Acquity UPLC Iclass system. For the HPLC system, HPLC–MS-grade ACN and MeOH (Fisher Solutions, El Cajon, USA) and deionized water (Milli-Q, 18.2 MΩcm^−1^) were used. ^68^Ga was obtained from a ^68^Ge/^68^Ga isotope generator (ITG, eluent: 0.05 M u.p. HCl). The cyclotron-produced ^44^Sc was obtained from Division of Nuclear Medicine, Department of Medical Imaging, University of Debrecen, Hungary. Activity measurements were carried out with a CAPINTEC CRC-15PET dose calibrator and a Perkin Elmer Packard Cobra gamma counter. Semipreparative RP HPLC, analytical HPLC and radio-HPLC were conducted using a Waters LC Module 1 HPLC and a Waters 2695 Alliance HPLC system, connected to UV detector and an ATOMKI CsI scintillation detector. Semipreparative RP HPLC purification was performed using a Luna C18 10 µm (250 × 10 mm) column; solvent A: 0.1% HCOOH; solvent B: acetonitrile. Analytical HPLC was performed using a Luna C18 3 µm (150 × 4.6 mm) column and a Kinetex C18 2.6 µm (100 × 4.6 mm) column; solvent A: oxalic acid (0.01 M pH = 3); solvent B: acetonitrile. Purification of labeled radiopharmaceuticals was carried out with the following chromatographic materials: SepPak^®^ C18 plus (Waters, Milford, MA, USA), LiChrolut EN (Merck, Darmstadt, Germany) and self-loaded DGA resin, 50–100 μm, 70 mg/cartridge (TrisKem). Mouse plasma was obtained from Sigma-Aldrich (Saint Lousis, MO, USA).

### 3.2. Chemistry

#### 3.2.1. Synthesis of 2-(2-Nitroimidazolyl) Ethyl-DO3AM (DO3AM-NI)

The synthesis of 2-(2-nitro-imidazol-1-yl) ethanamine was carried out by the method reported by Zha et al. [18]. To a mixture of 2-nitro-imidazole (1.55 g, 13.7 mmol) and *tert*-butyl 2-bromoethylcarbamate (4.2 g, 18.8 mmol) in dry dimethylformamide (20 mL), sodium iodide (0.2 g, 1.36 mmol) and potassium carbonate (3.79 g, 27.4 mmol) were added at room temperature. After 4 days stirring, the reaction mixture was diluted with ethyl acetate and washed with water. The organic layer was separated, dried and concentrated in vacuo. Crystallization of the residue from ethyl acetate gave *tert*-*N*-butyl 2-(2-nitroimidazolyl) ethylcarbamate (3.2 g, 91%) as pale-yellow crystals. To a solution of this compound (66 mg, 0.256 mmol) in dichloromethane (1 mL), trifluoracetic acid (1 mL) was added at room temperature. After stirring overnight, the reaction mixture was concentrated in vacuo. The crude product was used in the next step without further purification. 2-(2-nitro-imidazol-1-yl) ethanamine was conjugated to DOTA by using the reaction conditions, which was reported by Hoigebazar et al. [11]. To a solution of 2-(2-nitro-imidazol-1-yl) ethanamine (0.256 mmol) and DOTA (104 mg, 0.256 mmol) in a 1:1 mixture of *N,N*-dimethylformamide and water (3 mL), *N,N’*-dicyclohexylcarbodiimide (53 mg, 0.256 mmol) in 300 µL pyridine was added at room temperature. After stirring overnight, the reaction mixture was filtered, and the filtrate was concentrated in vacuo. The crude product was purified by semipreparative RP-HPLC. The semipreparative RP-HPLC was performed by using Luna C18 10 µm (250 × 10 mm) column with the following solvents: solvent A: 1% HCOOH, solvent B: acetonitrile, gradient: 0 min: 100% A, 2 min: 100% A, 32 min: 100% B, 40 min: 100% B 40.1 min: 100% A at a flow rate of 4 mL/min. The product was collected between 9.1 and 9.5 min, frozen and lyophilized to give DO3AM-NI (43 mg, 31%). HRMS ESI calc for: C_21_H_34_N_8_O_9_, 542.2449 [M]. Found: 543.2534 [M+H]^+^.

#### 3.2.2. pH-Potentiometric Studies

The equilibrium measurements were performed at constant ionic strength maintained by 0.15 M NaCl at 37 °C. For determining the protonation constants of DO3AM-NI, two parallel pH-potentiometric titrations were performed with 0.1611 M NaOH in a 5.00 cm^3^ sample containing the ligand at the concentration of 2.76 mM. The stability constant of Sc(DO3AM-NI) was assessed by ^1^H and ^45^Sc NMR in 0.7 mL “batch” samples containing the metal and the ligand at 3.32 mM and different concentrations of strong acid (0.05–0.20 M) aged 90 days (see below). The formation of protonated and ternary hydroxydo complex species was probed by pH-potentiometric titration by titrating the preformed complex (at a concentration of 1.95 mM) with standardized NaOH solution. For the calculation of the log*K*_MH-1L_ value, the V_NaOH_–pH data used were obtained in the pH range 1.68–11.85. The pH-potentiometric titrations were carried out with a Metrohm 785 DMP Titrino titration workstation with the use of a Metrohm-6.0233.100-combined electrode calibrated via a two-point calibration routine using KH-phthalate (pH = 4.005) and borax (pH = 9.177) buffers. The titrated samples were stirred with a magnetic stirrer, and N_2_ gas was bubbled through the solutions to avoid the effect of CO_2_. For the calculation of [H^+^] from the measured pH values, the method proposed by Irving et al. was applied [33]. For this, a 0.01M HCl solution was titrated with the standardized NaOH solution in the presence of 0.15 M NaCl ionic strength at 37 °C. The differences between the measured (pH_read_) and calculated pH (-log[H^+^]) values were used to obtain the equilibrium H^+^ concentration from the pH_meas_ values during the titrations. The ionic product of water (p*K*_w_) at 37 °C in 0.15 M NaCI was also calculated form these data and found to be p*K*w = 13.424. For the calculation of the equilibrium constants, the PSEQUAD program was used [34].

#### 3.2.3. ^1^H- and ^45^Sc-NMR Measurement

^1^H- and ^45^Sc-NMR measurements (400 MHz and 97 MHz, respectively) were performed with a Bruker DRX 400 (9.4 T) spectrometer equipped with a Bruker VT-1000 thermocontroller and a BB inverse z-gradient probe (5 mm). The kinetic experiments were performed at 25 °C, whereas the equilibrium studies were performed at 37 °C in aqueous solutions using D_2_O (in capillary) for locking. Typical parameters of ^45^Sc-NMR measurements used were pulse length p1 = 15.6 μs (90°), relaxation delay d1 = 0.1 s and 128 scans. The formation reaction of the Sc(DO3AM-NI) complex was followed by ^1^H- and ^45^Sc-NMR spectroscopy in a sample containing 3.80 mM ligand and 4.90 mM ScCl_3_ with a total acid concentration corresponding to a pH of 1.54. For the equilibrium studies, a total of six samples (0.70 cm^3^) were prepared with total H^+^ concentration of 0.05–0.2 M and [Sc(III)] = [DO3AM-NI] = 3.32 mM in water with constant ionic strength of 0.15 M NaCl. The Sc(III) exchange between the free Sc(III) and the Sc(DO3AM-NI) complex and the ligand exchange between the free and bound ligand were in the “slow exchange regime” on the actual NMR timescales (^45^Sc- and ^1^H NMR). Actually, the intensity of the ^45^Sc-NMR signal of Sc(III)_aq_ could precisely be measured while that of the broad signal corresponding to the Sc(DO3AM-NI) complex could not. Such integrations were performed for the ^1^H NMR signals of the 2-nitroimidazole moiety for the free ligand and that of the Sc(DO3AM-NI) complex. The molar integral values of the ^45^Sc-NMR signal of Sc(III)_aq_ were determined by recording the ^45^Sc-NMR spectra of a 50 mM ScCl_3_ solution containing 0.015 M HCl as a reference sample. Data sets included the NMR intensity values (3 values) and total concentrations (3 values), and the mass balance equations for the 7 samples were fitted by PSEQUAD program and gave the stability constant (log*K*_ScL_).

The dissociation kinetics of the Sc(DO3AM-NI) complex was probed in 1 M HCl solution by following the appearance and increase of ^45^Sc-NMR signal of Sc(III)_aq_ in the sample containing 7.30 mM Sc(DO3AM-NI) complex purified by HPLC.

### 3.3. Radiochemistry

#### 3.3.1. Radiolabeling DO3AM-NI with ^68^Ga^3+^

A ^68^Ge/^68^Ga-generator was eluted with 0.1 M aq. HCl, then the eluate containing ^68^Ga (100–120 MBq), was treated with the following post-processing method proposed by Eppard et al. [24]. ^68^Ga was trapped on the cation exchange resin (Strata SCX) and washed with 0.15 M HCl in an 8:2 mixture of ethanol and water (1 mL). The purified ^68^Ga was eluted with 0.9 M HCl in a 9:1 mixture of ethanol and water (3x100 µL). A fraction of 100 µL was transferred into an Eppendorf vial, then 60 µL of NaOAc/HOAc puffer (3M, pH = 4), 40 µL 5% NaOH, as well as 40 µL aq. stock solution of DO3A-NI (1 mg/mL) were added. The reaction mixture was kept at 95 °C for 15 min, then diluted with 1 mL water and passed through an SPE cartridge (SepPak^®^ C18 plus, Waters) preconditioned with 5 mL ethanol and 10 mL water. After purging of the cartridge with 1 mL of water, the labeled compound was eluted with a 1:1 mixture of ethanol and water (0.5 mL). The eluate was concentrated, then labeled product was dissolved in 100 µL of saline. The radiochemical purity of the product was determined with radio-HPLC on a Waters LC Module 1 HPLC with a Luna C18 3 µm (150 × 4.6 mm) column. Eluents: A = oxalic acid (0.01 M, pH = 3), B = acetonitrile, flow rate: 1 mL/min, gradient: 0 min: 100% A, 1 min: 100% A, 10 min: 100% B, 10.1 min: 100% A.

#### 3.3.2. Radiolabeling DO3AM-NI with ^44^Sc^3+^

A total of 120 mg of natural calcium (99.99%) was pressed into a pellet and pushed into the cavity of an aluminum target holder. Then, 60 min of irradiation with 30 μA beam current yielded approx. 300 MBq ^44^Sc. The irradiated Ca disc was dissolved in 3 M u.p. HCl (4 mL), and the solution was transferred onto a DGA cartridge preconditioned with 3 M u.p (3 mL) HCl, containing 70 mg resin. The cartridge was washed with 3 mL 3 M u.p. HCl and 3 mL 1 M HNO_3_ and eluted with 2 mL 0.1 M u.p. HCl in 200 µL fractions. The highest activity fractions were merged, and a fraction of 500 µL (100–150 MBq) was transformed into an Eppendorf vial, then 100 µL of NaOAc/HOAc puffer (3M, pH = 4), 20 µL 5% NaOH, as well as 5 µL aq. Stock solution of DO3AM-NI (1 mg/mL) were added. The reaction was performed at 95 °C for 15 min. Then, the reaction mixture was passed through a pre-conditioned SPE cartridge (LiChrolut EN, Merck). After purging of the cartridge with 1 mL of water, the labeled compound was eluted with a 1:1 mixture of ethanol and water (0.5 mL). The eluate was concentrated, and then labeled product was dissolved in 100 µL of saline. The radio-HPLC analysis was performed as described above using a Kinetex C18 2.6 µm (100 × 4.6 mm) column.

#### 3.3.3. Determination of logP Value of [^68^Ga]Ga(DO3AM-NI) and [^44^Sc]Sc(DO3AM-NI)

A 10 µL volume of [^68^Ga]Ga(DO3AM-NI) and [^44^Sc]Sc(DO3AM-NI) solution (~5 MBq) was mixed with 500 µL of 1-octanol and 490 µL of water in an Eppendorf vial, respectively. The mixture was shaken with a vortex shaker (600 rpm) for 10 min and centrifuged (6000 rpm) for 5 min. Then, 100 µL from the octanol phase and 1 μL from the aqueous phase were pipetted into vials, and the aqueous aliquot was diluted to 100 µL with water in order to minimize the effects of sample geometry and the high difference of activity concentrations in the two solvents. The radioactivity of the fractions was determined with a gamma counter. The measurements were performed in triplicates for both labeled compounds.

#### 3.3.4. Determination of [^68^Ga]Ga(DO3AM-NI) and [^44^Sc]Sc(DO3AM-NI) Stability in the Solution of Mouse Plasma, Na_2_H_2_EDTA and Oxalic Acid

A 10 µL volume of [^68^Ga]Ga(DO3AM-NI) and [^44^Sc]Sc(DO3AM-NI) solution was added to the solution of 90 µL of mouse plasma, Na_2_H_2_EDTA (0.01 M) and oxalic acid (0.01 M), respectively. Samples were analyzed by radio-HPLC at the beginning and after 60, 120 and 240 min. The analytical conditions were the same as detailed for the quality control of the labeled compound.

### 3.4. Biology

#### 3.4.1. Cell Culture

KB (human epidermal carcinoma) cell line was purchased from the American Type Culture Collection (ATCC; CCL-17™). Cells were cultured in Eagle’s Minimum Essential Medium (Sigma-Aldrich) supplemented with 10% fetal bovine serum (FBS) and 1% Antibiotic Antimycotic Solution (Sigma-Aldrich, Saint Louis, MO, USA). For tumor induction, the cells were used after 6–8 passages and 85% confluence. The viability of the cells was always higher than 90%, as assessed by the trypan blue exclusion test.

#### 3.4.2. Experimental Tumor Model

Adult male CB17 SCID mice (n = 25; from Charles River Laboratories by Innovo Kft., Hungary) were used at the age of 18 weeks. Mice were housed under sterile conditions in IVC cage system (Techniplast, Italy) at a temperature of 26 ± 3 °C, with 52 ± 10% humidity and artificial lighting with a circadian cycle of 12 h. Sterile semi-synthetic diet (Akronom Ltd., Budapest, Hungary) and sterile drinking water were available *ad libitum* to all the animals. Laboratory animals were kept and treated in compliance with all applicable sections of the Hungarian Laws and regulations of the European Union.

For tumor induction, SCID mice were injected with 5x10^6^ KB tumor cells in 0.9% NaCl (100 µL) subcutaneously into the left shoulder area. The tumor growth was assessed by caliper measurements, and tumor size was calculated using the following formula: (largest diameter × smallest diameter^2^)/2. In vivo experiments were carried out approximately 13 ± 1 days after intravenous injection of tumor cells at the tumor volume of approximately 110 mm^3^.

#### 3.4.3. In Vivo PET/MRI Imaging

For in vivo imaging studies, healthy control and KB tumor-bearing CB17 SCID mice were injected with 8.42 ± 0.38 MBq of ^68^Ga- or ^44^Sc-labeled DO3AM-NI via the lateral tail vein 13 ± 1 days after the inoculation of KB tumor cells. Then, 65 and 225 min after radiotracer injection, mice were anaesthetized by 3% isoflurane (Forane) with a dedicated small animal anesthesia device and whole-body T1-weighted MRI scans were performed (3D GRE EXT multi-FOV; TR/TE 15/2 ms; phase: 100; FOV 60 mm; NEX 2) using the preclinical *nanoScan* PET/MRI system with 1 Tesla magnetic field (Mediso Ltd., Hungary). After MRI imaging, 20 min static whole-body PET scans were acquired (90 min and 4 h after radiotracer injection).

#### 3.4.4. Quantitative PET Data Analysis

Quantitative radiotracer uptake was expressed in terms of standardized uptake value (SUV), SUV = [VOI activity (Bq/mL)]/[injected activity (Bq)/animal weight (g)], assuming a density of 1 g/mL. Volumes of interest (VOI) were manually drawn around the edge of the organ or tumor activity using the InterView™ FUSION (Mediso Ltd., Hungary) image analysis software. Tumor-to-muscle (T/M) ratios were computed as the ratio between the activity in the tumor VOI and in the background (muscle) VOI. Skeletal muscles of the right shoulder area were used as background.

#### 3.4.5. Ex Vivo Biodistribution Studies

For the determination of ex vivo biodistribution of the radiotracers, control and KB tumor-bearing mice were injected intravenously with 8.42 ± 0.38 MBq of ^68^Ga- or ^44^Sc-labeled DO3AM-NI via the tail vein. Mice were euthanized with 5% isoflurane 90 min or 4 h after intravenous radiotracer injection. Three tissue samples were taken from the selected organs, and their weight and radioactivity were measured with a calibrated gamma counter (Perkin-Elmer Packard Cobra, Waltham, MA, USA). The radiotracer uptake was expressed as %ID/g tissue.

#### 3.4.6. Data Analysis

Significance was calculated by two-way ANOVA, Mann–Whitney U-test and Student’s t-test (two-tailed). IBM SPSS Statistics and Microsoft Excel software were used for the statistical analysis. The significance level was set at *p* ≤ 0.05, unless otherwise indicated. Data are presented as mean ± SD of at least three independent experiments.

## 4. Conclusions

We have performed the physicochemical studies of the Sc(DO3AM-NI) complex, and the present work shows that substitution of an acetamide functionality for the acetate pendant arm on DOTA significantly changes the properties of the ligand. We found that the moderate decrease in stability and of the Sc(DO3AM-NI) complex together with the great inertness compared to the Sc(DOTA)^−^ complex clearly classify the DO3AM-NI ligand as an excellent Sc binder.

Furthermore, the radiolabeling of the DO3AM-NI ligand with a cyclotron produced ^44^Sc radiometal, which was successfully carried out with a high labeling yield and radiochemical purity. In addition, we compared the pharmacokinetic properties of [^44^Sc]Sc(DO3AM-NI) with the known hypoxia-specific [^68^Ga]Ga(DO3AM-NI) radiotracer by in vivo PET/MRI studies and ex vivo biodistribution studies. No significant difference was found between the tumor-specific accumulation of ^44^Sc- and ^68^Ga-labeled DO3AM-NI in KB tumors. However, higher accumulation of [^68^Ga]Ga(DO3AM-NI) was found in non-target tissues and organs compared to the accumulation of the ^44^Sc-labeled analog, resulting in a higher tumor-to-background ratio. Based on these results, we can conclude that the new [^44^Sc]Sc(DO3AM-NI) radiotracer is a promising molecular probe for PET imaging of tumor hypoxia due to its favorable features and high specificity.

## Figures and Tables

**Figure 1 pharmaceuticals-15-00666-f001:**
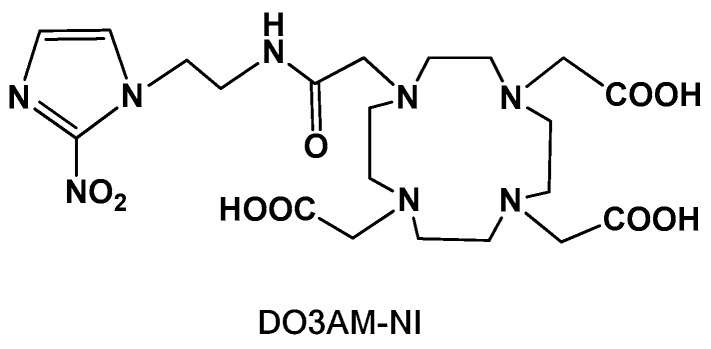
Chemical structure of the DO3AM-NI ligand.

**Figure 2 pharmaceuticals-15-00666-f002:**
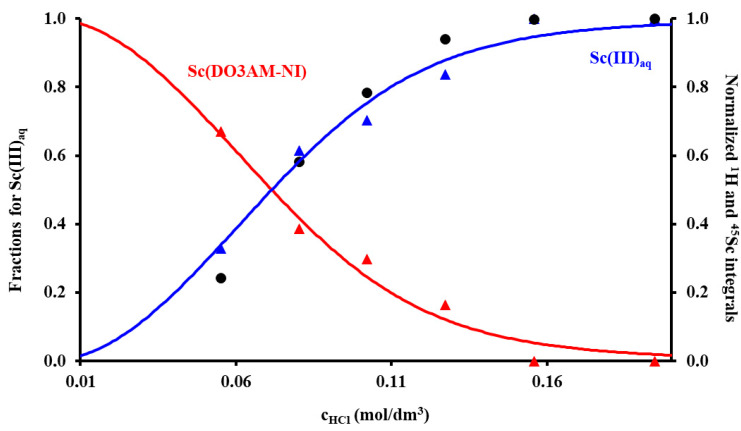
The species distribution of the Sc(III)–DO3AM-NI–H^+^ system (solid lines) calculated from the equilibrium data obtained by ^1^H and ^45^Sc-NMR spectroscopy co-plotted with the normalized intensity of the Sc(III)_aq_ (black dots) and those corresponding to the nitroimidazoyl protons in bound (red triangles) and free (blue triangles) ligand present in solution (3.32 mM ligand and ScCl_3_, *I* = 0.15 M NaCl, 37 °C).

**Figure 3 pharmaceuticals-15-00666-f003:**
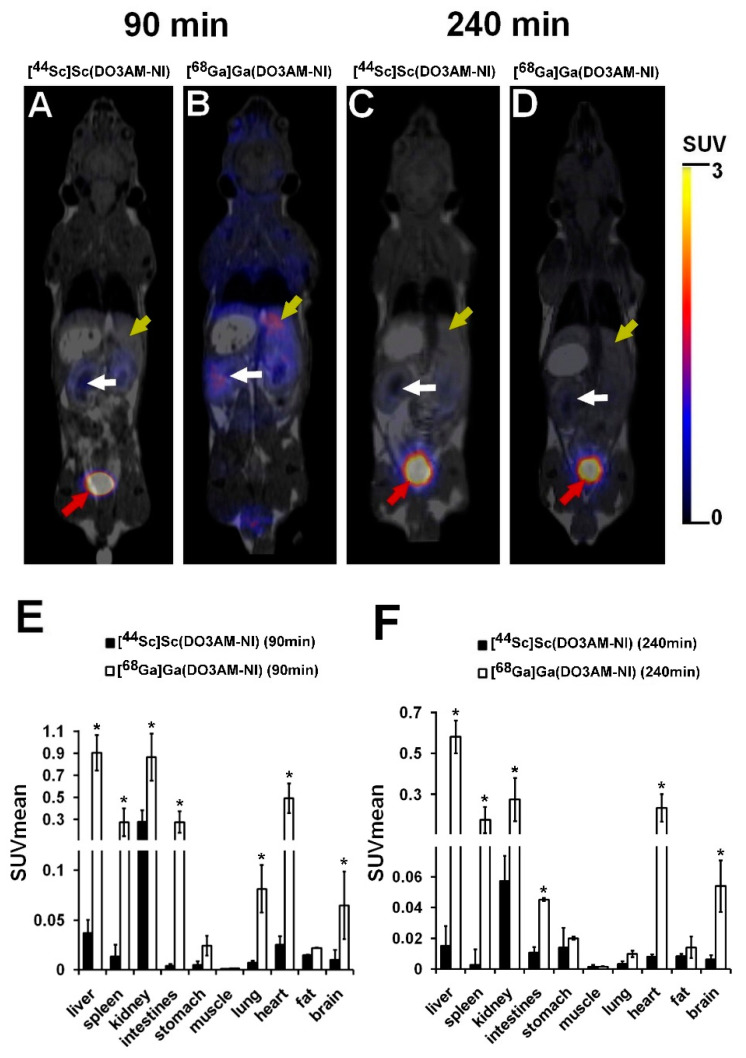
In vivo biodistribution of [^44^Sc]Sc(DO3AM-NI) and [^68^Ga]Ga(DO3AM-NI). Representative coronal PET/MRI images of healthy control SCID mice 90 and 240 min after intravenous injection of [^44^Sc]Sc(DO3AM-NI) (**A**,**C**) and [^68^Ga]Ga(DO3AM-NI) (**B**,**D**). Yellow arrows: liver; white arrows: kidney; red arrows: urinary bladder. Quantitative SUV analysis of in vivo biodistribution data (n = 5 control animals/radiotracer) 90 min (**E**) and 240 min (**F**) after tracer injection. SUV values are presented as mean ± SD. Significance level between [^44^Sc]Sc(DO3AM-NI) and [^68^Ga]Ga(DO3AM-NI): *p* ≤ 0.01 (*).

**Figure 4 pharmaceuticals-15-00666-f004:**
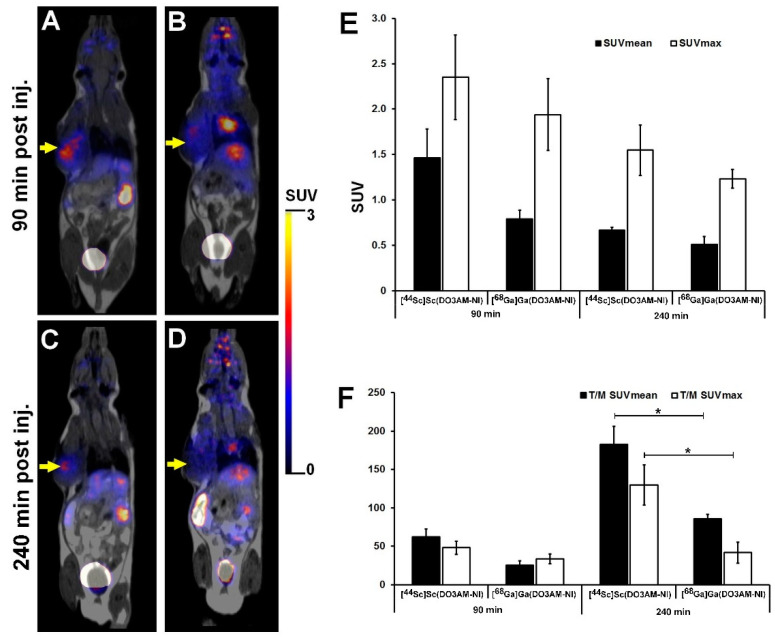
Representative in vivo whole-body PET/MRI imaging of KB tumor-bearing mice using [^44^Sc]Sc(DO3AM-NI) (**A**,**C**) and [^68^Ga]Ga(DO3AM-NI) (**B**,**D**) 90 and 240 min after intravenous tracer injection. Quantitative SUV analysis of [^44^Sc]Sc(DO3AM-NI) and [^68^Ga]Ga(DO3AM-NI) accumulation in experimental tumors (**E**,**F**). Decay-corrected PET/MRI images and data were obtained 13 ± 1 days after tumor cell inoculation. Yellow arrows: KB tumors. T/M: tumor-to-muscle ratio. Significance level: *p* ≤ 0.01 (*).

**Table 1 pharmaceuticals-15-00666-t001:** Stepwise protonation constants ^[a]^ and the sum of the protonation constants ^[b]^ of DO3AM-NI and DOTA, DO3AM^nBu^, DO3AM^NIM^, DTPA and AAZTA ligands used as comparative benchmarks, stability constants of their Sc(III) complexes ^[c]^ as well as *p*Sc ^[d]^ data.

Compound	DO3AM-NI	DOTA	DTPA	AAZTA	DO3AM^nBu^	DO3AM^NIM^
Condition	0.15 M NaCl, 37 °C	0.1 M Me_4_NCl, 25 °C ^[e]^	0.1 M Me_4_NCl, 25 °C ^[e]^	0.1 M KCl, 25 °C ^[f]^	1.0 M KCl, 25 °C ^[g]^	1.0 M KCl, 25 °C ^[g]^
log *K*_1_^H^	8.79(1)	12.90	10.65	11.26	9.78	10.17
log *K*_2_^H^	8.59(1)	9.72	8.66	6.62	9.05	9.02
log *K*_3_^H^	4.23(1)	4.62	4.36	3.86	4.53	4.41
log *K*_4_^H^	2.49(1)	4.15	2.82	2.45	3.17	2.94
log *K*_5_^H^	1.00(4)	2.29	2.03	1.88	2.19	1.99
log *K*_6_^H^	n.d.	1.34	1.31	1.46	n.d.	n.d.
log *K*_7_^H^	n.d.	n.d.	1.30	n.d.	n.d.	n.d.
**log *β*_015_**	**25.10**	**33.68**	**28.52**	**26.04**	**28.72**	**28.53**
**log *K*_Sc•L_**	**22.36(4)**	**30.79**	**27.43**	**27.69**	**n.a.**	**n.a.**
log *K*_ScL•H_	n.d.	1.00	1.36	0.86	n.a.	n.a.
log *K*_ScL•OH_	10.88(5)	n.d.	12.44	n.d.	n.a.	n.a.
***p*Sc**	**20.74**	**23.92**	**23.88**	**24.72**	**n.a.**	**n.a.**

^[a]^ *K*_i_^H^ = [H_i_L]/[H^+^]·[H_i-1_L]; ^[b]^ log *β*_015_ = **∑**log*K*_5_^H^; ^[c]^
*K*_ScL_ = [ScL]/[Sc^3+^][L], the protonation constant *K*_ScLH_ = [ScLH]/[ScL][H^+^] and the ternary hydroxydo species *K*_ScLOH_ = [ScLOH][H^+^]/[ScL]; ^[d]^ *p*Sc values were calculated near to physiological conditions: *p*Sc =-log[Sc(III)]_free_, at [Sc(III)]_tot_ = 1 × 10^−8^ M, [L]_tot_ = 1 × 10^−7^ M and pH = 7.4); therefore, under these conditions in the absence of any complexation the minimum value of *pSc* is 8.0; n.d.: not detected; n.a.: not aquired; ^[e]^ Ref. [19]; ^[f]^ Ref. [20]; ^[g]^ Ref. [21].

**Table 2 pharmaceuticals-15-00666-t002:** Ex vivo biodistribution of [^44^Sc]Sc(DO3AM-NI) and [^68^Ga]Ga(DO3AM-NI) (%ID/g) in healthy control SCID mice 90 and 240 min after tracer injection. Significance level between [^44^Sc]Sc(DO3AM-NI) and [^68^Ga]Ga(DO3AM-NI) at 90 and 240 min: *p* ≤ 0.01 (*). %ID/g values are presented as mean ± SD.

	[^44^Sc]Sc(DO3AM-NI)		[^68^Ga]Ga(DO3AM-NI)	
Organ/Tissue	90 min (n = 3)	240 min (n = 5)	90 min (n = 5)	240 min (n = 5)
Blood	0.08 ± 0.01	0.03 ± 0.01	0.81 ± 0.22 *	0.65 ± 0.11 *
Urine	57.78 ± 12.41	6.23 ± 1.25	65.94 ± 5.21	17.08 ± 7.97 *
Liver	0.14 ± 0.09	0.20 ± 0.04	0.43 ± 0.11 *	0.37 ± 0.10 *
Spleen	0.07 ± 0.02	0.06 ± 0.01	0.29 ± 0.14 *	0.32 ± 0.18 *
Kidney	1.17 ± 0.41	0.89 ± 0.23	1.32 ± 0.47	1.21 ± 0.17
Small intestine	0.10 ± 0.05	0.03 ± 0.01	0.25 ± 0.10 *	0.14 ± 0.04 *
Large intestine	0.05 ± 0.01	0.04 ± 0.004	0.20 ± 0.13 *	0.15 ± 0.06 *
Stomach	0.09 ± 0.03	0.03 ± 0.005	0.21 ± 0.04 *	0.22 ± 0.06 *
Muscle	0.01 ± 0.01	0.007 ± 0.002	0.10 ± 0.03 *	0.06 ± 0.03 *
Lung	0.10 ± 0.04	0.04 ± 0.003	0.49 ± 0.08 *	0.46 ± 0.09 *
Heart	0.04 ± 0.01	0.03 ± 0.01	0.26 ± 0.06 *	0.22 ± 0.04 *
Fat	0.01 ± 0.001	0.01 ± 0.01	0.13 ± 0.10 *	0.12 ± 0.08 *

**Table 3 pharmaceuticals-15-00666-t003:** Ex vivo evaluation of [^44^Sc]Sc(DO3AM-NI) and [^68^Ga]Ga(DO3AM-NI) uptake (%ID/g) in KB tumors 13 ± 1 days after subcutaneous tumor induction. Significance level between the tumor-to-muscle ratios (T/M) of [^44^Sc]Sc(DO3AM-NI) and [^68^Ga]Ga(DO3AM-NI) at 90 and 240 min: *p* ≤ 0.01 (*). %ID/g values are presented as mean ± SD.

	[^44^Sc]Sc(DO3AM-NI)		[^68^Ga]Ga(DO3AM-NI)	
Tumor	90 min	240 min	90 min	240 min
**KB tumor (n = 5)**	0.82 ± 0.11	0.70 ± 0.14	0.62 ± 0.10	0.46 ± 0.07
**T/M**	79.30 ± 5.26 *	99.35 ± 5.14 *	6.28 ± 1.11	7.51 ± 1.13

## Data Availability

All data are contained in this article and its related Appendix A.

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
