# Peer review of "Synthesis, Physicochemical, Labeling and In Vivo Characterization of ^44^Sc-Labeled DO3AM-NI as a Hypoxia-Sensitive PET Probe"

_pharmaceuticals, 2022, doi:10.3390/ph15060666_

Round 1
Reviewer 1 Report
- Y. Liu, J. Zhang, Y. Guo, P.Wang, Y. Su, X. Jin, X. Zhu, C. Zhang. . Exploration, 2022, 2, 20210172.
- Chu T, Rice E J, Booth G T, et al. Chromatin run-on and sequencing maps the transcriptional regulatory landscape of glioblastoma multiforme[J]. Nature genetics, 2018, 50(11): 1553-1564.
- Brennan C W, Verhaak R G W, McKenna A, et al. The somatic genomic landscape of glioblastoma[J]. Cell, 2013, 155(2): 462-477.
- Y. Zheng, Y. Han, Q. Sun, Z.Li, Exploration, 2022,2,20210166. https://doi.org/10.1002/EXP.20210166.
- Frattini V, Trifonov V, Chan J M, et al. The integrated landscape of driver genomic alterations in glioblastoma[J]. Nature genetics, 2013, 45(10): 1141-1149.
Reviewer 2 Report
Szücs and colleagues investigated a DO3AM-based peptide labeled with 44Sc for hypoxia PET imaging. The authors described the synthesis of the peptide, its radiolabeling to 44Sc, and performed a comprehensive in vivo and in vitro analysis of the radiotracer. In vivo comparison of [44Sc]Sc(DO3AM-NI) and [68Ga]Ga(DO3AM-NI) was performed.
The manuscript is well-written and the experiments are comprehensively described.
Specific comments
- A few spelling and grammatical errors are scattered across the manuscript. Please read through carefully and correct them.
- Introduction: “In addition, this positron emitter radionuclide has several advantages over the widely used 68Ga isotope (t1/2 = 68 min, I = 89%, Emax (β+) = 1.92 MeV) [13].” Please provide similar information for 44Sc as provided here for 68
- Introduction: “To the best of our knowledge this study is the first report of the preparation and in vivo evaluation of 44Sc-labeled 2-nitroimidazole-based tracer for tumor hypoxia imaging.” This statement is not necessary here. It suffices to mention that the labeling of 2-nitroimidazole to 44Sc and the evaluation of the utility of the radiotracer for PET hypoxia imaging have not been previously reported.
- Methods: “Tumour-to-muscle (T/M) ratios were computed as the ratio between the activity in the tumor VOI and in the background (muscle) VOI.” Please specify the muscle used to compute background activity.
- Methods: Please specify the statistical software used for data analysis.
- Results – section 2.1.3 Radiochemistry: This section contains a lot of information that should be in the Methods. Furthermore, not much is reported here with respect to the results of the radiochemistry experiments performed by the authors. Please correct.
- Figure 3E: Please adjust the scale of the y-axis so that the smaller bars are prominent and can be easily seen. The taller bars can be truncated to accommodate this change.
- Clinical 68Ga-based PET imaging is commonly acquired after a 60-minute uptake period. Please provide a justification for your choice of 90 minutes for the early time-point comparison of [44Sc]Sc(DO3AM-NI) and [68Ga]Ga(DO3AM-NI).
- Nitroimidazole is a target for hypoxia and not necessarily a probe for tumor detection. What are the indicators that hypoxia was present in the tumors of the animals used in the in vivo experiments?
- Figure 4 legend: For the sake of consistency, please express time in minutes (i.e. 240 minutes) rather than hours.
- The conclusion is rather too long. The conclusion should be a distillation of the salient findings from the study not an opportunity for another round of discussion as done here. Please correct.
Reviewer 3 Report
An in vivo comparison of 44Sc-labeled DO3AM-NI with its known 68Ga-labeled analog for tumor hypoxia using PET is described, demonstrating the utility of [44Sc]DO3AM-NI) molecular probe for PET-MRI imaging of tumor hypoxia.
The study is well performed, methods clearly described and correctly concluded. A carefully revision of English is recommended for more clarity.
Reviewer 4 Report
On request of Pharmaceuticals I revised the paper entitled “Synthesis, physicochemical, labeling and in vivo characterization of a DO3AM-based hypoxia sensitive 44Sc-labeled PET probe” by Dániel Szücs et al. The paper deals with the synthesis of a new radiotracer for tumor hypoxia imaging. The Authors prepared the nitroimidazolyl derivative of DOTA according to previous studies and performed its radiolabeling with 68Ga and 44Sc isotope. The work is well described and written and the scientific significance is good as the radiolabeling with 44Sc has never been performed before. In vivo and ex vivo biodistributions in mice were also studied and confirmed the suitability of this new tracer for PET imaging of tumor hypoxia. However before publication some adjustments are required.
Specific comments:
- The abstract is too general in the section of the description of the results. It should be more detailed i.e. highlighting the differences in specific tissue accumulation
- Figure 3 is not clear. The color of the legend at 240 min makes it difficult to see the differences between the two sample types. It is practically impossible to understand which column refers to 44Sc(DO3AM-NI) and to 68Ga(DO3AM-NI) 240 min. The Authors should provide two graphics, one for 90 min and another for 240 min. In this way the results will be more clear. Also the statistical analysis is not clear. The asterisks refer to what time? Please provide a new analysis comparing the data corresponding to the same time point.
Round 2
Reviewer 2 Report
Thank you for your response. My comments have been addressed. I have no further comments.
Reviewer 4 Report
No more comments are needed.